# OpenReview forum: "Enhancing Cross-Lingual and Cross-Domain Adaptability in Large Language Models for Software Engineering"
_ICLR.cc/2025/Conference — Submitted to ICLR 2025_

### Official Review · Reviewer_EAc1 · 2024-10-23

**Soundness:** 2
**Presentation:** 2
**Contribution:** 2
**Rating:** 5
**Confidence:** 3

**Summary:**

This paper introduces UDA-EDCM (Unsupervised Domain Adaptation - Enhanced Dynamic Code Modeling), a new framework designed to address cross-lingual and cross-domain adaptability in code modeling. The authors propose a measure-theoretic approach for domain adaptation combined with Riemannian manifold-based feature space alignment. The paper introduces novel theoretical tools, including the Geodesic Flow Kernel, and presents a unified performance bound for UDA-EDCM. Empirical results demonstrate significant improvements over existing models in zero-shot and few-shot learning for programming languages, highlighting UDA-EDCM’s potential in advancing code intelligence.

**Strengths:**

1. The paper presents a thorough mathematical treatment, introducing new concepts such as the Geodesic Flow Kernel and a unified bound on UDA performance.
2. The use of differential geometry and optimal transport theory in domain adaptation is a novel and creative combination, providing a fresh perspective on feature space alignment in code modeling.
3. The UDA-EDCM framework is well-rounded, incorporating both theoretical analysis and empirical validation. The multi-modal approach (handling both natural and programming languages) is particularly impressive.
4. The experimental results, especially in few-shot and zero-shot settings, clearly demonstrate the framework's efficacy across a variety of programming languages.

**Weaknesses:**

1. The paper is difficult to follow in some places due to the dense mathematical formalism. Concepts like the Geodesic Flow Kernel and operator-theoretic analysis, while innovative, are presented with minimal intuitive explanations. More concrete examples and visualizations would help.
2. The use of differential geometry and spectral analysis introduces significant complexity, which may hinder practical adoption. While the theoretical results are elegant, the applicability of these advanced techniques in real-world, large-scale software engineering environments is not sufficiently addressed. The computational overhead introduced by Riemannian manifold calculations and optimal transport may outweigh the performance gains in practical applications, especially when working with large datasets or resource-constrained systems.
3. The experimental evaluation is comprehensive in terms of comparing the UDA-EDCM framework with other models on several tasks, but it lacks diversity in the types of programming languages and paradigms tested. While the paper demonstrates performance across multiple languages, it does not evaluate the framework's adaptability to more niche or specialized programming paradigms (e.g., functional, logic, or domain-specific languages). This limits the generalizability of the framework in complex, real-world codebases.
4. The scalability of UDA-EDCM to industrial-scale codebases is not fully discussed. Many modern software projects involve millions of lines of code with highly varied styles, legacy code, and mixed programming paradigms. The experiments presented focus on smaller-scale tasks, but how well the proposed framework scales in terms of both memory and processing time is not clear. There is a need to test the approach on significantly larger datasets or long-term maintenance tasks involving evolving codebases.
5. The paper does not provide sufficient ablation studies to isolate the contributions of individual components, such as the Geodesic Flow Kernel or the information-theoretic bounds. Since UDA-EDCM is a complex framework with multiple innovations, understanding which parts contribute the most to performance is crucial for practical adoption. Including ablation studies would make the experimental results more convincing by demonstrating the impact of each component.

**Questions:**

1. Could the authors provide more practical insights on how the UDA-EDCM framework can be integrated into existing software engineering pipelines? Are there specific use cases where this framework would be particularly beneficial?
2. The paper claims to handle cross-lingual tasks effectively, but how does the model perform with truly low-resource programming languages? Are there any specific limitations in such scenarios?
3. The framework relies on complex mathematical operations, including Riemannian geometry and functional analysis. How does this affect the model's computational efficiency, particularly in large-scale, real-world deployments?

---

### Official Review · Reviewer_s7hy · 2024-11-01

**Soundness:** 1
**Presentation:** 1
**Contribution:** 2
**Rating:** 1
**Confidence:** 3

**Summary:**

This paper provides a theoretical framework for domain adaptation in cross-lingual and cross-domain code modeling, with empirical evidence.

**Strengths:**

Domain adaptation is a hot topic and relevant the ICLR community. Further theoretical understanding of this problem could greatly benefit both researchers and practitioners. The authors make bold theoretical claims.

**Weaknesses:**

- The presentation of the paper is weak. There are large white spaces all over the paper (bullet points, equations, figures). All the images are very small and hard to read. Figures 2, 3, and 4 overlap with the figure description, making the description hard to read. Figures are referred to as Tables in the text. The poor formatting of the paper makes it feel incomplete.

- The paper should provide intuition for its theoretical results. As it reads now, the implication of each theorem is missing. Instead, the authors simply state the theorem followed by a proof. I would suggest placing the proofs in the Appendix so that the main paper can focus on the "why?" for each theorem.

- It is unclear what the proposed algorithm is. I read through the paper multiple times, but it seems Section 3 talks about theoretical results and Section 4 immediately jumps into experimental results without any explanation of what UDA-EDCM is. Furthermore, the only Figure explaining the algorithm, Figure 1, is not referenced anywhere in the text.

- The experimental evidence is not clearly explained. The authors simply list the performance improvements in Section 4.3 and 4.4, with no explanation why the performance is improved.

- The authors should provide more explanation why they do not compare against any general-purpose large language models such as CodeLlama, Llama, GPT, DeepSeekCoder, Claude, or Mistral.

**Questions:**

See Weaknesses.

---

### Official Review · Reviewer_ny1V · 2024-11-02

**Soundness:** 2
**Presentation:** 1
**Contribution:** 2
**Rating:** 3
**Confidence:** 4

**Summary:**

This paper introduces UDA-EDCM, a framework that revolutionizes Unsupervised Domain Adaptation (UDA) for code intelligence. UDA-EDCM leverages mathematical concepts from measure theory, differential geometry, and information geometry to address the challenges of adapting AI systems to new coding environments. The framework introduces a novel measure-theoretic formulation of domain adaptation, utilizes optimal transport theory to minimize domain discrepancy, and employs a Riemannian manifold approach for feature space alignment.

**Strengths:**

- UDA-EDCM is built upon the theoretical foundations of measure theory, differential geometry, and information geometry, providing rigorous guarantees for its performance and convergence.
- The paper introduces novel theoretical tools such as the geodesic flow kernel and a unified performance bound for UDA-EDCM, offering new insights and methods for cross-domain adaptation.

**Weaknesses:**

- The paper lacks an in-depth discussion of the complexity of software engineering. a) Limitations in code representation: The paper mentions that UDA-EDCM uses the geodesic flow kernel to capture the intrinsic geometric structure of code representations, but the representation of code itself may have limitations. For example, abstract concepts, control flow, and dependency relationships in code are difficult to represent with simple vectors, which may affect the effectiveness of domain adaptation. b) Dynamic code structure: The structure of code evolves with version changes, such as code refactoring and evolution. How does UDA-EDCM handle this dynamism in software engineering and ensure the adaptability of the model across different code versions?
- This work uses information geometry methods for model optimization, but this approach may have limitations. For instance, the computation of the Fisher information matrix can be complex and computationally intensive, affecting the efficiency of model training. The paper lacks discussion and experiments on computational complexity.
- I am concerned about the match between theory and experiment. The experiments provided in the article are too sparse to verify the following: Can the theoretical bounds proposed in the article accurately predict the actual performance of the UDA-EDCM model? Do the theoretical assumptions proposed in the article, such as the Wasserstein distance, fully capture the differences between the source and target domains? Can the feature space used in the article effectively capture the semantics and structure of the code?
- The readability of the article is poor, especially in the experimental section.

**Questions:**

- Significant differences may exist between code paradigms, such as object-oriented and functional programming. How does UDA-EDCM handle cross-paradigm transfer and ensure the adaptability of the model across different paradigms?

---

### Official Review · Reviewer_9xbS · 2024-11-05

**Soundness:** 2
**Presentation:** 1
**Contribution:** 2
**Rating:** 3
**Confidence:** 4

**Summary:**

This paper introduces the Enhanced Dynamic Code Modeling (UDA-EDCM) system, which employs mathematical concepts to tackle the complexities of diverse programming and natural languages. The paper provides some theoretical proofs to show the feasibility of using UDA in the context of code modeling.

**Strengths:**

N/A

**Weaknesses:**

* The paper is not well written. It seems there are no details about how UDA-EDCM works. The theorem part is mainly on showing the feasibility of using UDA in the context of code modeling. Also, there's no reference to the definition and theorems. It is not clear which part of the derivation comes from the paper itself.
* The related work section is unfortunately short, with only a high-level discussion on recent code LLMs.
* In the last Section 5, the paper concludes the introduction of "AdaptiCode-ML". However, this is the first time that this term has been brought up. There seem to be multiple typos to distinguish between "Figure" and "Table". Also, there's no discussion about Figure 3.
* There is no discussion about "Figure 1: Schematic Representation of the Multi-Modal Code Adaptation Framework" and it is not sure if this is the UDA-EDCM referred to by the authors.

**Questions:**

Please refer to "Weakness" section.

---

### Official Review · Reviewer_eYx8 · 2024-11-06

**Soundness:** 2
**Presentation:** 2
**Contribution:** 2
**Rating:** 5
**Confidence:** 3

**Summary:**

The paper introduces UDA-EDCM, a framework for unsupervised domain adaptation (UDA) in cross-lingual and cross-domain code modeling, by incorporating advanced mathematical techniques like measure theory, differential geometry, and information theory. The proposed method leverages optimal transport theory for feature space alignment and Riemannian manifolds for seamless domain adaptation. The key components of the proposed method include Domain-Adaptive Context-Aware Code Modeling (DACACM) and Dynamic Code Environment Generation (DCEG), which enable context-aware and dynamic adaptations.

**Strengths:**

- The proposed method involves robust mathematical foundations, employing measure theory, differential geometry, and information theory to effectively address domain discrepancies.

- The proposed method demonstrates strong zero-shot and few-shot learning capabilities.

**Weaknesses:**

- The quality of the figures, such as figure 2, 3, 4, should be improved.

- The experiments are insufficient. The proposed method is only evaluated on text conversion and text-to-code (code-to-text) generation tasks. It would strengthen the paper to include experiments on more general software engineering tasks, such as program repair and fault localization. Additionally, the authors should consider testing with more recent and advanced Large Language Models, such as Llama 3.1, which is pre-trained across multiple languages.

- The computational costs of high-dimensional operations in real-world settings are unclear. Is there any analysis and discussion on this aspect?

**Questions:**

See above.

---

### Meta-Review · Area_Chair_XP3G · 2024-12-15

**Metareview:**

The authors did not submit the response to address reviewers' concern. In the internal discussion, no member from the review team would like to champion this paper. Below are my person comments.

**Research Question**

This paper considers the unsupervised domain adaptation problem in the context of cross-lingual and cross-domain code modeling.

**Challenge Analysis**

In general, the targeted research question is a well-defined one. The authors fail to demonstrate what the specific challenges are in the context of cross-lingual and cross-domain code modeling. Why the current unsupervised domain adaptation methods cannot tackle this problem?

**Philosophy**

Since the targeted challenges are missing, the philosophy is missing, either.

**Techniques**

The authors provided their solutions without any rationality.

**Experiments**

The experiments are not solid. The authors should clearly illustrate the setup (datasets, baseline methods, evaluation metric).

**Presentation**

It is strongly suggested to improve the presentation to deliver a logical and smooth reading experience. Moreover, the visualization and format should be improved as well.

**Additional Comments On Reviewer Discussion:**

No objection from reviewers who participated in the internal discussion was raised against the reject recommendation.

---

### Decision · Program_Chairs · 2025-01-22

Reject